# Understanding Embedding Scaling in Collaborative Filtering

**Yicheng He**  *yh84@uiuc.edu*
*University of Illinois Urbana-Champaign*

**Kaiyu Zhou**  *kaiyu001@e.ntu.edu.sg*
*Nanyang Technological University*

**Haoyue Bai**  *baihaoyue621@gmail.com*
*Arizona State University*

**Fengbin Zhu**  *zhfengbin@gmail.com*
*National University of Singapore*

**Yonghui Yang**[*]  *yh_yang@nus.edu.sg*
*National University of Singapore*

**Reviewed on OpenReview:** *https://openreview.net/forum?id=3f5HtLqnaY*

## Abstract

Scaling recommendation models into large recommendation models has become one of the most widely discussed topics. Recent efforts focus on components beyond the scaling embedding dimension, as it is believed that scaling embedding may lead to performance degradation. Although there have been some initial observations on embedding, the root cause of their non-scalability remains unclear. Moreover, whether performance degradation occurs across different types of models and datasets is still an unexplored area. Regarding the effect of embedding dimensions on performance, we conduct large-scale experiments across 10 datasets with varying sparsity levels and scales, using 4 representative classical architectures. We surprisingly observe two novel phenomena: *double-peak* and *logarithmic*. For the former, as the embedding dimension increases, performance first improves, then declines, rises again, and eventually drops. For the latter, it exhibits a perfect logarithmic curve. Our contributions are threefold. First, we discover two novel phenomena when scaling collaborative filtering models. Second, we gain an understanding of the underlying causes of the double-peak phenomenon. Lastly, we theoretically analyze the noise robustness of collaborative filtering models, with results matching empirical observations.

## 1 Introduction

Scaling up Transformer (Vaswani, 2017) parameters to boost performance in NLP tasks is a key reason behind the success of large language models. Therefore, a natural question arises: can we replicate similar success in collaborative filtering? It is widely agreed that embedding dimension is one of the key controllable parameters in collaborative filtering. However, there is also a consensus that scaling the embedding dimension may leads to performance degradation (Guo et al., 2024; Ardalani et al., 2022; Zhang et al., 2024; Ye et al., 2025; Xu et al., 2025; Peng et al., 2025). For example, Guo et al. (2024) argues: "*Worsely, increasing the embedding dimension does not sufficiently improve the performance or even hurts the model*", and Zhang et al. (2024) believes: "*...Merely expanding the sparse component of a model does not enhance its ability to capture the complex interactions ...*". As a result, recent works (Zhang et al., 2024; Guo et al., 2024) on scaling recommendation models bypass the seemingly unviable path of expanding embedding dimension.

---

[*]Corresponding author

A further question arises: why do scaling embedding dimensions fail to improve recommendation performance? Guo et al. (2024) observe that when the embedding dimensions are scaled, embedding will collapse. Specifically, they conduct a spectral analysis based on singular value decomposition of the learned embedding matrices and find that most of the singular values are extremely small. Furthermore, several studies on efficient recommendation Liu et al. (2020); Qu et al. (2023); Wang et al. (2024); Luo et al. (2024); Liu et al. (2021) treat the embedding dimensionality as a learnable parameter and optimize it through predefined objectives, thereby enhancing the model's overall effectiveness. Because, they think that there exists an optimal dimensionality for embeddings. In other words, performance exhibits a single-peak phenomenon as the embedding is scaled. Overall, previous works observed that increasing the dimensionality not only worsens performance but also causes changes in the singular values of the embedding.

In this paper, we are more interested in understanding why the performance deteriorates with scaling and what the underlying causes are. Would it follow the previously observed single-peak phenomenon?

To enable a comprehensive investigation, we conducted large-scale experiments across 10 datasets with varying sizes and sparsity levels (Table 3), using 4 representative collaborative filtering models (reason at D.4). Surprisingly, we observe two novel phenomena: *double-peak* and *logarithmic* (Figure 1), apart from the common single-peak. Additionally, we have two observations, for the same model, some datasets exhibit a double-peak trend, while others show a logarithmic increase. More interestingly, we also find that even within the same dataset, the phenomena are not always consistent. For example, BPR (Rendle et al., 2009) exhibits a double-peak pattern, while SGL (Wu et al., 2021) follows a logarithmic trend.

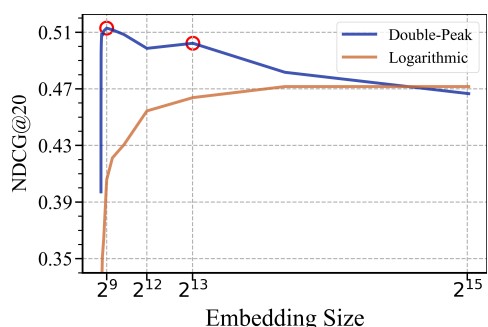

Figure 1: Double-peak: first rising, then falling, followed by another rise and decline; logarithmic, performance follows a logarithmic increase.

Clearly, we prefer the logarithmic scaling for its expected stable performance, while the emergence of the double-peak phenomenon indicates a limitation in scalability. Since both phenomena appear across different datasets within the same model, we boldly speculate that this is due to certain negative factors in the interactions. Inspired by He et al. (2022), we concluded that the issue originates from the noise present in the data, and we conducted a thorough analysis of the causes of noise at each stage. We also conducted an in-depth analysis of the noise resistance and robustness of various collaborative filtering models and the results are generally consistent with the empirical evidence.

**Our contributions are summarized as follows**.

- To the best of our knowledge, our work is the first report of the double-peak and logarithmic phenomenon. In the realm of collaborative filtering, there has long been a perplexing question: why does expanding embedding dimensions not always result in performance enhancements? Our research make further additions to it.

- Through in-depth analysis, we identified interaction noise, in addition to overfitting, as the key factors causing the double-peak phenomenon. To test this idea, we propose a simple yet effective sample drop strategy that helps collaborative filtering models scale more effectively. For instance, the embedding dimensions can be expanded up to 32,768 without significant risk of collapse.

- We theoretically analyzed the noise robustness of BPR, NeuMF, LightGCN, and SGL, with results largely aligning with empirical observations, indirectly validating our analysis. For instance, BPR shows poor noise robustness and more frequently exhibits the double-peak phenomenon, while SGL is more robust and tends to follow a logarithmic pattern.

**The implications for future research**.

- Transformer-based LLMs have garnered significant attention. Most of researches (Zhang et al., 2024; Xu et al., 2024) aim to bring the success of the Transformer architecture into recommendation models to

achieve superior scaling performance. However, in this paper, we explore a different path: finding the "Transformer" within collaborative filtering models. We discovered that SGL possesses structural advantages in this regard. Therefore, future research directions may focus on better filtering clean interactions and further scaling up SGL for enhanced performance.

## 2 Background: Collaborative Filtering

**Notations**. We use lowercase letters for scalars. Denote the set of real numbers by $\mathbb{R}$. We use bolded lowercase letters $\mathbf{p}$ to represent the entire embedding, and subscripts to refer to specific embedding $\mathbf{p}_u$. We use calligraphic letters e.g., $\mathcal{N}$ to denote the value spaces (other notions in Appendix A).

**User and Item Embedding**. In collaborative filtering, the core components involve representing users and items using embedding vectors. The user embedding contains the user's preference information, which can better capture the similarity between different users. Similarly, the item embedding contains the item's feature information. Let us define: $\mathbf{p}_u \in \mathbb{R}^m$ as the embedding vector for user $u$, where $u = 1, 2, \ldots, m$. $\mathbf{q}_i \in \mathbb{R}^n$ as the embedding vector for item $i$, where $i = 1, 2, \ldots, n$. $m$ is the dimensionality of the embedding.

**Collaborative Filtering**. To compute the preference score $\hat{r}_{ij}$ of user $u$ for item $i$, many powerful and representative collaborative filtering models have been proposed, such as BPR (Rendle et al., 2009), NeuMF (He et al., 2017), LightGCN (He et al., 2020) and SGL (Wu et al., 2021).

❶ BPR (Rendle et al., 2009) calculates the preference score as the inner product of user and item embeddings, or employs a neural network to capture nonlinear interactions:

$$\hat{r}_{ij} = \mathbf{p}_u^\top \mathbf{q}_i = \sum_{l=1}^{k} u_{il} \cdot v_{jl}, \quad \hat{r}_{ij} = f_{\mathrm{MLP}}(\mathbf{p}_u, \mathbf{q}_i),$$

where $u_{il}$ and $v_{jl}$ are the $l$-th components of $\mathbf{p}_u$ and $\mathbf{q}_i$, respectively.

❷ NeuMF (He et al., 2017) is a representative deep learning-based collaborative filtering model with a classic two-tower architecture. It fuses Generalized Matrix Factorization (GMF) and Multi-Layer Perceptron (MLP) to capture both linear and nonlinear user–item interactions:

$$\hat{r}_{ij}^{\mathrm{GMF}} = \mathbf{p}_u^\top \mathbf{q}_i, \quad \hat{r}_{ij}^{\mathrm{MLP}} = f_{\mathrm{MLP}}([\mathbf{p}_u \| \mathbf{q}_i]),$$

where $[\cdot \| \cdot]$ denotes vector concatenation. The final prediction $\hat{r}_{ij}$ is computed by combining the outputs from GMF and MLP, typically through a fully connected layer.

❸ LightGCN (He et al., 2020) simplifies GCN by removing feature transformation and nonlinear activation to better capture high-order connectivity in user-item graphs. It adopts symmetric normalization in the graph convolution process, aggregating embeddings from neighboring nodes using a degree-aware weighting scheme:

$$\mathbf{p}_u^{(l+1)} = \sum_{i \in \mathcal{N}_u} \frac{1}{\sqrt{|\mathcal{N}_u||\mathcal{N}_i|}} \mathbf{q}_i^{(l)}, \quad \mathbf{q}_i^{(l+1)} = \sum_{u \in \mathcal{N}_i} \frac{1}{\sqrt{|\mathcal{N}_i||\mathcal{N}_u|}} \mathbf{p}_u^{(l)}.$$

This symmetric normalization allows the model to more accurately propagate information and better capture user interests.

❹ SGL (Wu et al., 2021) introduces self-supervised learning by augmenting graph structure and node features, incorporating contrastive objectives to improve robustness and alleviate sparsity:

$$\mathcal{L} = \underbrace{\mathcal{L}_{\mathrm{BPR}}}_{\text{collaborative filtering}} + \gamma \underbrace{\mathcal{L}_{\mathrm{cont}}}_{\text{contrastive}},$$

where $\mathcal{L}_{\mathrm{BPR}}$ is the Bayesian Personalized Ranking loss $-\sum_{(u,i,j) \in \mathcal{D}} \ln \sigma\left(\mathbf{p}_u^\top \mathbf{q}_i - \mathbf{p}_u^\top \mathbf{q}_j\right)$ and $\mathcal{L}_{\mathrm{contrastive}}$ is the contrastive loss encouraging consistency between augmented views. In addition, $\gamma$ controls the contrastive task's contribution.

# 3 Analysis of Scaling: Scale it up! Scale it up again!

We focus on three points: First, what strategy we use to increase embedding size. Second, what phenomenon we observe in experiments. Third, why these phenomena happen.

## 3.1 Observation Settings

Our experiments are based on the RecBole [1] (Xu et al., 2023). We select four popular collaborative filtering models: BPR (Rendle et al., 2009), NeuMF (He et al., 2017), LightGCN (He et al., 2020) and SGL (Wu et al., 2021), along with ten classic datasets, covering different level of scale and sparsity. These datasets are described in the Appendix 3. To ensure a thorough investigation, our experiments were conducted on a setup equipped with 8*80G GPUs. Other settings regarding all model parameters, optimizer, learning rate, etc. are in the Appendix D.1.

## 3.2 Double-peak and Logarithmic Phenomenon

In the past understanding, when the embedding dimension increases, the performance shows a single-peak phenomenon of first increasing and then decreasing. However, we believe that this understanding lacks microscopic and extensive observations. Through large-scale experiments (Figure 2), we find that on the ML-100K dataset, when the embedding dimension increases, the overall performance of the BPR model is indeed consistent with previous studies, showing an overall trend of first increasing and then decreasing. However, we discover more details for the first time: as the embedding dimension increases, the recommendation performance initially increases significantly, then decreases, then increases slightly, and finally decreases again. We call this new trend ***double-peak phenomenon*** (as shown in Figure 2 (b, c, e, f)). Moreover, we noticed that the most of peak performance occurs at $2^9$, which are different from the traditional settings of 32, 64, and 128. But unexpectedly, in LightGCN and SGL, this phenomenon is alleviated a lot, and SGL alleviates it better than LightGCN (as shown in Figure 2 (k, h)). In addition, our experiments show that on the Modcloth dataset, BPR, NeuMF, LightGCN, and SGL all show a continuous upward trend. For example, even when the dimension is set to $2^{14}$, the performance of LightGCN is still rising. We call this ideal trend the ***logarithmic phenomenon*** (as shown in Figure 2 (a)), that is, as the embedding dimension increases, the performance continues to improve in a stable and predictable manner, while the marginal gains gradually diminish. Such a logarithmic trend is particularly desirable, as it provides consistent performance improvements with increasing embedding size. The logarithmic phenomenon completely exceeds the past understanding. Compared with the NDCG@20 at 128 dimensions, the performance has increased by 25.57%. On the contrary, if we only set the dimension to 128, we will lose 34.36% of the potential performance improvement. Currently, most research in the recommendation field, such as debiasing (Chen et al., 2021; 2023) and denoising (Wang et al., 2021; He et al., 2024), believes that a 10% performance improvement is meaningful. However, in our observations, simply increasing the dimension can achieve a performance improvement far exceeding that of elaborately designed methods.

To ensure the generality of the phenomenon, we design a series of experiments, conducting observations across other 7 datasets (E.1) and also analysis the results in detail. Empirical results confirm that the phenomenon is indeed widespread.

**Two Observations**. ❶ The same model exhibits different phenomenons across different datasets. As shown in Figure 2 (a, d), BPR exhibits a double-peak phenomenon on the ML-100K dataset, while showing logarithmic growth on the Modcloth dataset. This seems to be due to some negative factor in ML-100K that causes scaling to fail. ❷ The same dataset exhibits different phenomena across different models. As shown in Figure 2 (a, b, c), we observe that BPR exhibits a double-peak phenomenon, while LightGCN and SGL show logarithmic growth. Moreover, we believe this might be because the structures of LightGCN and SGL incorporate mechanisms that mitigate negative factors. Moreover, the logarithmic growth of SGL appears to be more ideal.

---

[1]https://www.recbole.io/docs/index.html

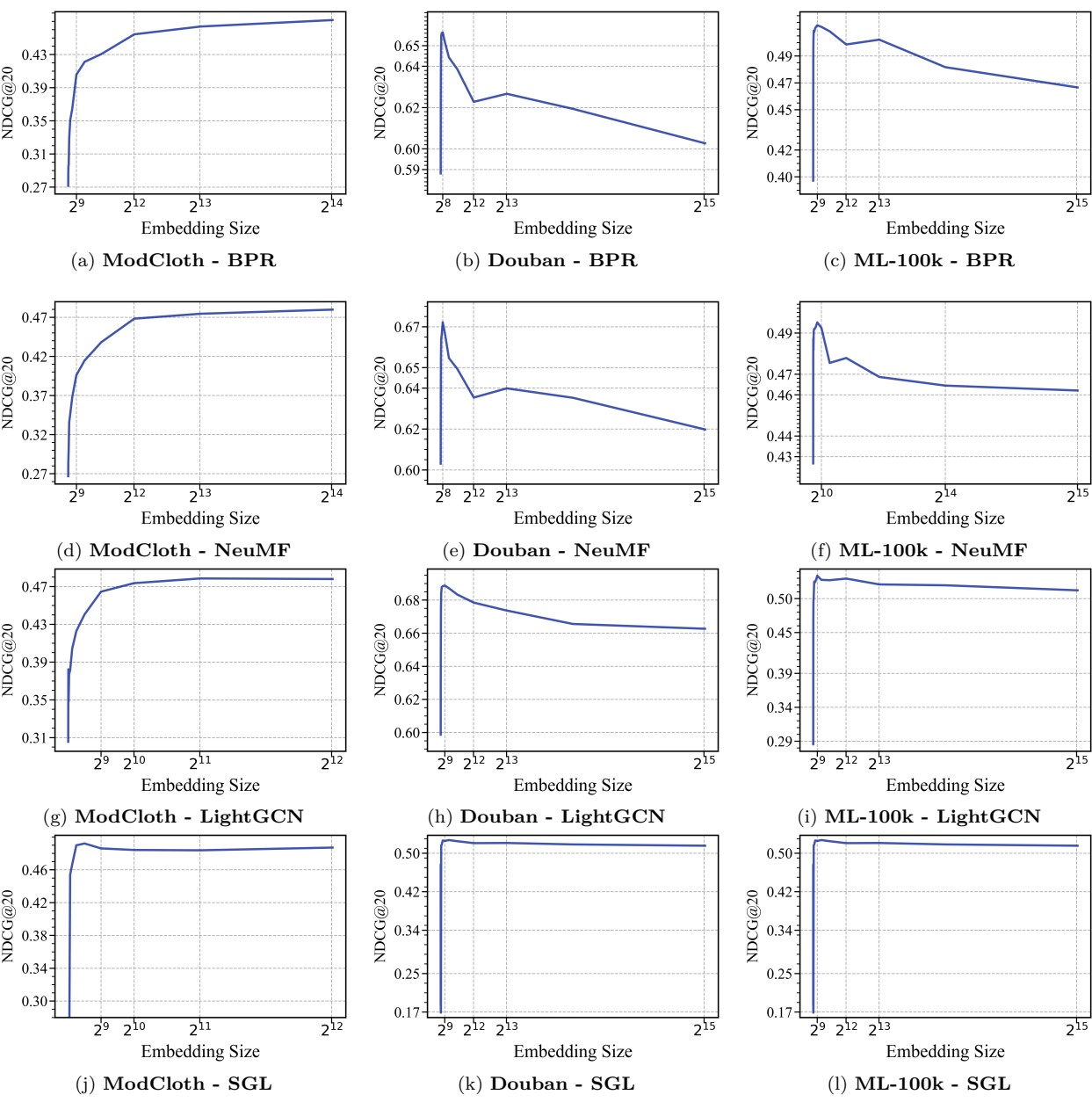

Figure 2: Scale the embedding dimension exponentially by a factor of 2 across different collaborative filtering models and datasets. Each row corresponds to a model (BPR, LightGCN, SGL, NeuMF), and each column represents a dataset (Modcloth, Douban, ML-100k).

## 3.3 Why Two Phenomena Occur

Based on the previous two observations, we believe this is due to the presence of certain negative factors in the dataset that BPR and NeuMF cannot resist, while LightGCN and SGL can. When these negative factors are mitigated, a logarithmic trend emerges, whereas if influenced by these factors, a double-peak pattern is observed. In addition, the difficult-to-understand aspect of the double-peak phenomenon lies in the third stage, where performance rises again. If this stage were absent, with performance simply rising and then falling, it could easily be explained as overfitting. However, the appearance of the third stage prevents us from attributing it to overfitting, and we believe this is the key to understanding this phenomenon. For

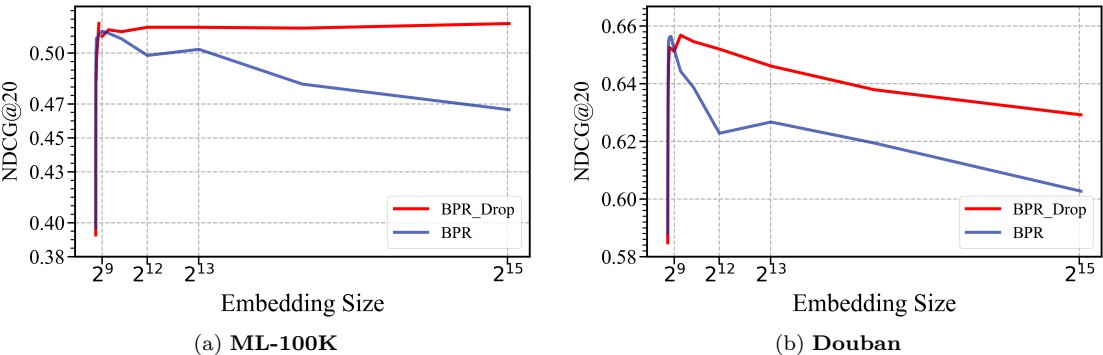

Figure 3: Comparing the standard BPR with the denoising strategy-based BPR_Drop in ML-100K and Douban, we can clearly observe that the performance degradation has significantly improved.

logarithmic phenomenon, this is an ideal trend. If every model can exhibit this trend on every dataset, it would be wonderful.

Upon close examination of Figures 2, 4, 5, we observe that SGL tends to exhibit such a logarithmic trend. We attribute this behavior to the contrastive learning framework in SGL, which provides both regularization and robustness enhancement. In contrast, other methods lack such a mechanism, which may explain their deviation from this desirable pattern.

**Noisy Interactions Induces the Double-peak Phenomenon**. For the double-peak phenomenon, this is a novel phenomenon. Our approach to analyzing this phenomenon was inspired by two phenomena in machine learning: the memorization effect on noisy labels (Zhang et al., 2017) and the sparse double descent (He et al., 2022) which found that the sparsity of model parameters and the level of label noise jointly affect the model's performance (more about noisy interactions at B.3).

**Definition 1 (Sparse Double Descent)** *(He et al., 2022) is a phenomenon that occurs during model pruning. It is characterized by the model's test performance first decreasing due to overfitting as sparsity increases, then improving once overfitting is alleviated after reaching a certain level of sparsity, and finally decreasing again as sparsity increases, due to the loss of useful information.*

We think that scaling the embedding dimension is related to the sparsification of neural networks in recommendation tasks, as increasing the dimension effectively reduces the sparsity of the neural network. Overall, this process can be summarized into four phases: the model first tends to learn clean samples (Phase 1, improves), then memorizes noisy samples (Phase 2, declines). As the embedding size increases, the enlarged space dilutes the negative impact of noise (Phase 3, rebounds); finally, an excessively large embedding size leads to overfitting (Phase 4, drops). Notably, the resulting double-peak curve resembles the phenomenon of *sparse double descent* (He et al., 2022), since both are driven by sparsity and data noise. Nevertheless, there are important distinctions: sparse double descent is typically studied in general machine learning settings involving label noise and model pruning, whereas our double-peak phenomenon arises specifically in collaborative filtering when scaling the embedding dimension under noisy interactions. Specifically, *in the first stage*, the clean learning stage, the model initially learns from clean interactions. As the dimensionality of the embedding increases, the model has more space to accommodate user interests, leading to performance improvements. *In the second stage*, the noise memorization stage, the presence of interaction noise, combined with a dimensionality large enough to accommodate noisy interactions, causes the model to begin learning from these noisy interactions, resulting in performance degradation. *In the third stage*, the sweet stage, the embedding space enlarges to adapt to interaction noise, thereby enhancing the model's robustness. *In the fourth stage*, the overfitting stage, the continuous increase in embedding dimensions leads to a higher fit to the training dataset, which subsequently reduces generalization performance.

## 4 One of the Keys to Scaling: Improve Robustness to Noisy Interactions

In this section, we conduct a detailed analysis based on the models and explore the characteristics of each model, focusing on three main questions: Why are BPR and NeuMF easily affected by noise? Why does graph collaborative filtering exhibit noise resistance? Why does graph self-supervised learning perform better than graph collaborative filtering?

### 4.1 Analysis of the Resistance to Noise of BPR

We first analyze why the BPR model is particularly sensitive to noisy interactions. Our analysis is based on the degradation of representation quality under noisy preference distributions.

**Definition 2 (Representation Quality)** *Let $\Theta^* = \{\mathbf{p}_u^*, \mathbf{q}_i^*\}$ denote the ideal latent representations learned from clean data $\mathcal{D}_0$, and $\Theta = \{\mathbf{p}_u, \mathbf{q}_i\}$ be any learned parameters. The representation quality is defined as:*

$$Q(\Theta) := \|\Theta - \Theta^*\|_F^2 = \sum_u \|\mathbf{p}_u - \mathbf{p}_u^*\|_2^2 + \sum_i \|\mathbf{q}_i - \mathbf{q}_i^*\|_2^2. \tag{1}$$

**Theorem 1 (Representation Perturbation Under Noisy Interactions)** *Let $\Theta_0$ be the minimizer of the clean empirical loss $\mathbb{E}_{x \sim \mathcal{D}_0}[\ell(\Theta; x)]$, and consider a noisy interaction distribution $\mathcal{D}_\delta := (1 - \delta)\mathcal{D}_0 + \delta\mathcal{N}$, where $\mathcal{N}$ is a noise distribution and $\delta \in [0, 1]$ is the noise ratio. Define the noise gradient shift as $\Delta_{\mathrm{noise}} := \mathbb{E}_{x \sim \mathcal{N}}[\nabla_\Theta \ell(\Theta_0; x)] - \mathbb{E}_{x \sim \mathcal{D}_0}[\nabla_\Theta \ell(\Theta_0; x)]$. Assume the loss $\ell(\Theta; x)$ is twice differentiable and the expected clean loss is locally strongly convex around $\Theta_0$. Then, for sufficiently small $\delta$, the representation quality satisfies:*

$$Q(\Theta_\delta) := \|\Theta_\delta - \Theta_0\|_F^2 \le \delta^2 \cdot \|H^{-1}\Delta_{\mathrm{noise}}\|_F^2 + o(\delta^2), \tag{2}$$

*where $\Theta_\delta$ is the minimizer under $\mathcal{D}_\delta$, and $H := \nabla_\Theta^2 \mathbb{E}_{x \sim \mathcal{D}_0}[\ell(\Theta; x)]|_{\Theta=\Theta_0}$ is the Hessian at the clean optimum. The complete proof can be found in C.1*

**Remark 1** *Theorem 1 reveals that the degradation in representation quality under noisy interactions is quadratically proportional to the noise ratio $\delta$, and linearly controlled by $\|\Delta_{\mathrm{noise}}\|$, which is dictated by the model architecture.*

**Lemma 1 (High Gradient Sensitivity in BPR)** *In BPR, the scoring function is defined as $f(u, i, j) = \mathbf{p}_u^\top(\mathbf{q}_i - \mathbf{q}_j)$, which is linear in its parameters. Therefore, the per-sample gradient is:*

$$\nabla_\Theta \ell(\Theta; x) \propto (\mathbf{q}_j - \mathbf{q}_i, \mathbf{p}_u, -\mathbf{p}_u), \tag{3}$$

*with unbounded norm when embeddings grow. As a result, $\|\Delta_{\mathrm{noise}}\|$ tends to be large, making BPR more vulnerable to noisy samples.*

**Remark 2** *This explains the double-peak phenomenon observed in BPR training, especially under large embedding dimensions, due to overfitting to noise.*

**Experimental Evidence**. To validate the robustness of BPR, we adopted a simple strategy: we applied denoising collaborative filtering methods to observe whether there would be an improvement in performance, particularly when the embedding dimensions are high (Code at D.2).

$$\Theta^{t+1} \leftarrow \Theta^t - \eta \nabla \ell_{\mathrm{BPR}}\left(\Theta^t; \{x_i \mid \mathcal{L}(x_i) \le \tau\}\right). \tag{4}$$

This equation illustrates a dynamic sample-dropping strategy (Wang et al., 2021) to enhance BPR's noise robustness. During gradient updates, only samples with losses below threshold $\tau$ are retained, filtering out high-loss interactions likely to be noise. Experiments show this approach effectively mitigates noise in high-dimensional embedding spaces while improving recommendation performance (as shown in Figure 3). The threshold generally depends on the dropout rate: higher dropout leads to a lower threshold, and vice versa. We conducted extra experiments on Figure 7 with different dropout rates for further evidence.

### 4.2 Analysis of the Resistance to Noise of NeuMF

We now analyze the noise sensitivity of NeuMF from a Jacobian-based perspective. Despite the inclusion of nonlinear components, the gradient propagation through multiple nonlinear layers can amplify noise, depending on the Jacobian of the model output with respect to the parameters.

**Lemma 2 (Noise-Induced Gradient Amplification)** *Let $f(u,i)$ be the NeuMF prediction function, and let $\Theta \in \mathbb{R}^d$ denote all model parameters. Define the Jacobian matrix of the scoring function with respect to parameters as $J_x := \nabla_\Theta f(u,i) \in \mathbb{R}^{1 \times d}$ for input $x = (u,i)$. Then for a noisy interaction $x \sim \mathcal{N}$, the norm of the loss gradient is bounded by:*

$$\|\nabla_\Theta \ell(\Theta; x)\| \leq L_\ell \cdot \|J_x\|, \tag{5}$$

*where $L_\ell$ is the Lipschitz constant of the loss function. Hence, the gradient norm is directly governed by the spectral norm $\|J_x\|$ of the Jacobian.*

**Theorem 2 (Gradient Instability in NeuMF)** *Let $f(u,i) = \text{MLP}([\mathbf{p}_u; \mathbf{q}_i]) + \mathbf{p}_u^\top \mathbf{q}_i$ be the NeuMF scoring function, where $\text{MLP}$ has $L$ layers with weight matrices $W^{(1)}, \ldots, W^{(L)}$, each with spectral norm $\|W^{(l)}\|_2$. Then the Jacobian satisfies:*

$$\|J_x\| = \left\| \frac{\partial f(u,i)}{\partial \Theta} \right\| \leq \left( \prod_{l=1}^{L} \|W^{(l)}\|_2 \right) \cdot \|\nabla_z \ell(f(u,i), y)\|, \tag{6}$$

*which grows multiplicatively with depth, implying NeuMF can have exponentially amplified gradient sensitivity under noisy interactions. The proof precess are at C.2.*

**Corollary 1 (Comparative Noise Sensitivity: NeuMF vs. BPR)** *Although BPR has unbounded inner product gradients w.r.t. embeddings, its Jacobian norm is linear in embedding size, i.e., $\|J_x^{\text{BPR}}\| \propto \|\mathbf{p}_u\| + \|\mathbf{q}_i\|$. In contrast, NeuMF's Jacobian grows with the product of layer spectral norms, potentially leading to*

$$\|J_x^{\text{NeuMF}}\| > \|J_x^{\text{BPR}}\|, \tag{7}$$

*in deep or poorly regularized MLPs. Hence, NeuMF may not exhibit superior robustness, and may in fact be more susceptible to gradient perturbations from noise.*

### 4.3 Analysis of the Resistance Noise of Graph Collaborative Filtering

LightGCN exhibits a relatively expected logarithmic growth pattern. We believe this is likely due to the inherent noise-resistant structure of LightGCN. We think that the process of aggregating information from a user's neighboring nodes in graph convolution (He et al., 2020) is highly consistent with Mixup (Zhang et al., 2018).

$$(\tilde{\mathbf{u}}, \tilde{\mathbf{v}}) = \underbrace{\left( \frac{1}{|\mathcal{N}_i|} \sum_{k \in \mathcal{N}_i} \mathbf{u}_k, \mathbf{v}_i \right)}_{graph\ convolution};$$

$$(\tilde{\mathbf{u}}, \tilde{\mathbf{v}}) = \underbrace{\left( \sum_{i=1}^{N} \lambda_i \mathbf{u}_i, \sum_{i=1}^{N} \lambda_i \mathbf{v}_i \right)}_{Mixup}, \text{ s.t. } \sum_{i=1}^{N} \lambda_i = 1. \tag{8}$$

Specifically, first-order neighbors aggregate information about items, while second-order neighbors aggregate information from other users. Since Mixup is regarded as noise-resistant, we hypothesize that graph-based collaborative filtering possesses similar noise-resistant capabilities. Moreover, this idea aligns with findings

in Han et al. (2024).

$$
\begin{aligned}
(\tilde{\mathbf{u}}, \tilde{\mathbf{v}}) &= \left( \frac{1}{|\mathcal{N}_i|} \sum_{k \in \mathcal{N}_i} \mathbf{u}_k, \mathbf{v}_i \right) \\
&= \left( \sum_{k \in \mathcal{N}_i} \frac{1}{|\mathcal{N}_i|} \mathbf{u}_k, \sum_{k \in \mathcal{N}_i} \frac{1}{|\mathcal{N}_i|} \mathbf{v}_i \right) \\
&\overset{\lambda_i = \frac{1}{|\mathcal{N}_i|}}{=} \left( \sum_{k \in \mathcal{N}_i} \lambda_i \mathbf{u}_k, \sum_{k \in \mathcal{N}_i} \lambda_i \mathbf{v}_i \right).
\end{aligned}
\tag{9}
$$

This makes it easy to understand why LightGCN is noise-resistant, as Mixup, being a classic and popular method for domain generalization, is effective in mitigating noise disturbances. Next, we analyze from the spectral perspective.

**Definition 3 (Spectral Representation of Graph Convolution)** *Let the adjacency matrix of the user-item interaction graph $\mathcal{G}$ be $A \in \mathbb{R}^{(m+n) \times (m+n)}$, and the degree matrix be $D$. The normalized adjacency matrix is $\tilde{A} = D^{-1/2} A D^{-1/2}$. Performing eigendecomposition on $\tilde{A}$:*

$$
\tilde{A} = U \Lambda U^T, \quad \Lambda = diag(\lambda_1, \lambda_2, \ldots, \lambda_{m+n}),
\tag{10}
$$

*where $|\lambda_1| \geq |\lambda_2| \geq \cdots \geq |\lambda_{m+n}|$, and $U$ is the orthogonal eigenvector matrix.*

**Definition 4 (Low-Rank Property of Graph Convolution)** *The embedding propagation in LightGCN (Eq.2) can be represented as a spectral filtering process. After $L$ layers of propagation, the final user/item embeddings are:*

$$
\mathbf{p}_u = \sum_{l=0}^{L} \beta_l \tilde{A}^l \mathbf{p}_u^{(0)}, \quad \mathbf{q}_i = \sum_{l=0}^{L} \beta_l \tilde{A}^l \mathbf{q}_i^{(0)},
\tag{11}
$$

*where $\beta_l = \frac{1}{L+1}$. In the spectral domain, this is equivalent to applying a low-pass filter to the initial embeddings:*

$$
\mathbf{p}_u = U \left( \sum_{l=0}^{L} \beta_l \Lambda^l \right) U^T \mathbf{p}_u^{(0)}.
\tag{12}
$$

*The filter coefficient matrix $\sum_{l=0}^{L} \beta_l \Lambda^l$ attenuates high-frequency components (small $|\lambda_k|$) while preserving low-frequency components (large $|\lambda_k|$), leading to low-rank embeddings. The complete proof can be found in C.4.*

### 4.4 Analysis of the Resistance Noise of Self-supervised Graph Collaborative Filtering

Self-supervised graph collaborative filtering (SGL) is a subset of graph-based collaborative filtering. Naturally, we can also infer that self-supervised graph collaborative filtering inherently possesses the ability to resist interaction noise due to the graph convolution mechanism. Moreover, in most cases, SGL outperforms LightGCN. So, what additional module plays a crucial role in this difference?

SGL adopts self-supervised contrastive learning. SGL generates multi-view representations via stochastic graph perturbations and optimizes contrastive loss. For view generation, we create augmented graphs $\mathcal{G}'$ and $\mathcal{G}''$ by: edge dropout: Remove edges with probability $\rho$ and feature masking: zero out embedding dimensions with probability $\rho$. For contrastive loss, we maximize consistency between views for the same node:

$$
\mathcal{L}_{\text{cont}} = -\sum_{u \in \mathcal{U}} \log \frac{\exp\left(\text{sim}(\mathbf{z}_u', \mathbf{z}_u'')/\tau\right)}{\sum_{v \in \mathcal{U}} \exp\left(\text{sim}(\mathbf{z}_u', \mathbf{z}_v'')/\tau\right)},
\tag{13}
$$

where $\text{sim}(\cdot)$ is cosine similarity, $\tau$ is a temperature hyperparameter, and $\mathbf{z}_u', \mathbf{z}_u''$ are augmented embeddings of $u$.

**Definition 5 (Mutual Information Objective in Contrastive Learning)** *SGL maximizes the mutual information $I(z'_u; z''_u)$ between augmented views $z'_u$ and $z''_u$ via the contrastive loss $\mathcal{L}_{cont}$. By the information bottleneck principle:*

$$I(z'_u; z''_u) = H(z'_u) - H(z'_u|z''_u), \tag{14}$$

*where $H(\cdot)$ denotes entropy. Minimizing $\mathcal{L}_{cont}$ forces $z'_u$ and $z''_u$ to share noise-invariant information.*

**Theorem 3 (Embedding Space Constraint via Contrastive Learning)** *Let the noise perturbation $\delta_u$ satisfy $\delta_u \perp \mathcal{S}$, where $\mathcal{S}$ is the subspace of clean interaction signals. The contrastive loss in SGL implicitly constrains the embedding vectors $z_u$ to lie within $\mathcal{S}$:*

$$\forall u, \quad \|Proj_{\mathcal{S}^\perp}(z_u)\|_2 \leq \epsilon, \tag{15}$$

*where $\epsilon$ is a constant related to noise level, and $\mathcal{S}^\perp$ is the orthogonal complement of $\mathcal{S}$. The complete proof can be found in C.3.*

**Corollary 2 (Noise Filtering in SGL)** *As demonstrated in Theorem 3, SGL achieves noise resistance by employing low-pass filtering in graph convolution to attenuate high-frequency noise while retaining essential low-frequency components, and by imposing a subspace constraint in contrastive learning to further reduce noise variance and improve noise robustness. Mathematically, SGL embeddings are:*

$$z_u = \underbrace{Proj_{\mathcal{S}}(z_u^*)}_{low\text{-}rank\ signal} + \underbrace{Proj_{\mathcal{S}^\perp}(\delta_u)}_{noise}, \tag{16}$$

*where $\|Proj_{\mathcal{S}^\perp}(\delta_u)\|_2 \leq \epsilon$, and $\epsilon$ decreases with contrastive learning strength $\gamma$.*

## 5   Discussion

**The quality of the recommendation dataset is more important than we thought**. The quality of the dataset, especially noisy interactions, yields a substantial influence on the scaling recommendation performance. When the dataset contains less noise, the model can scale more smoothly, with algorithms adapting and expanding more efficiently to handle larger volumes of data. Conversely, a high level of noise data muddles the underlying patterns, undermining the model's ability to scale effectively and leading to suboptimal performance as the scale increases.

**Collaborative filtering models with noise-resistant structures have more advantages when it comes to scaling**. We can clearly see that, compared to BPR, NeuMF and LightGCN, SGL demonstrates stronger scaling capabilities. Even when the embedding dimensions are significantly increased, SGL maintains stable performance or stays close to peak performance, unlike BPR, which experiences a sharp decline. This advantage stems from the inherent noise resistance of SGL's structure. We believe that further exploration of this structure could potentially lead to the emergence of a "Transformer" in the field of collaborative filtering.

## 6   Conclusion and Limitations

**Conclusion**. Our contributions lie in the discovery of two new phenomena (double-peak and logarithmic), the interpretation of their causes, and the theoretical analysis of noise robustness in representative collaborative filtering models. Specifically, to investigate whether scaling embedding dimensions can improve collaborative filtering models, we conducted extensive observational experiments across 10 datasets and 4 representative models. Compared to the previously recognized single-peak phenomenon, we identified two novel phenomena: double-peak and logarithmic. We conducted a detailed analysis of the underlying cause of the double-peak effect and found it to result from noise interactions. To empirically validate our hypothesis, we introduced a denoising loss function, which supported our explanation. Furthermore, we provided a theoretical analysis of the models' robustness to noise with results aligned with our empirical observations.

**Limitations**. Due to limitations in available computational resources, we were unable to conduct experiments on a broader range of collaborative filtering models. Additionally, our observations were based

primarily on the Top@K metric with K set to 20. With sufficient resources, future work could explore additional values of K (e.g., 5, 10) and incorporate more fine-grained evaluation metrics to further validate the generality of the observed phenomena.

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

# Appendix of "Understanding Embedding Scaling"

**Contents**

## A Notions

Table 1: Symbol Definitions

| Symbol | Description |
| --- | --- |
| $u, v$ | Indices representing users. |
| $i, j$ | Indices representing items. |
| $\mathbf{p}_u \in \mathbb{R}^k$ | Latent representation vector for user $u$. |
| $\mathbf{q}_i \in \mathbb{R}^k$ | Latent representation vector for item $i$. |
| $\Theta = \{\mathbf{p}_u, \mathbf{q}_i\}$ | Set of all user and item latent representations. |
| $\Theta^* = \{\mathbf{p}_u^*, \mathbf{q}_i^*\}$ | Ideal set of user and item latent representations satisfying true preference relations. |
| $\Theta_0$ | Optimal representations by minimizing the loss function without noise. |
| $\Theta_\eta$ | Optimal representations by minimizing the loss function with noise proportion $\eta$. |
| $\mathcal{D}_0$ | Set of authentic (noise-free) interaction triples $(u, i, j)$. |
| $\mathcal{D}_\eta$ | Set of noisy interaction triples $(u, j, i)$. |
| $\mathcal{D} = \mathcal{D}_0 \cup \mathcal{D}_\eta$ | Combined set of authentic and noisy interaction triples. |
| $\eta \in (0, 1)$ | Proportion of noisy interactions in the dataset. |
| $\lambda > 0$ | Regularization parameter in the loss function. |
| $\sigma(z) = \frac{1}{1+e^{-z}}$ | Sigmoid function. |
| $\mathcal{L}_0(\Theta)$ | Loss function without noise. |
| $\mathcal{L}_\eta(\Theta)$ | Loss function component due to noisy interactions. |
| $\mathcal{L}(\Theta)$ | Total loss function with noise $= \mathcal{L}_0(\Theta) + \mathcal{L}_\eta(\Theta)$. |
| $Q(\Theta)$ | Representation quality metric. |
| $\|\cdot\|_F$ | Frobenius norm. |
| $\|\cdot\|_2$ | Euclidean (L2) norm. |
| $\mu > 0$ | Strong convexity parameter. |
| $c > 0$ | Lower bound constant related to noisy interactions. |

## B Related Work

### B.1 Scaling Recommendation.

Large language models (LLM) have shone brightly in today's era, and one of the key reasons is that the performance of the foundational Transformer (Vaswani, 2017) model can scale with parameter expansion. Recently, a few efforts (Ardalani et al., 2022; Zhang et al., 2023; 2024; Guo et al., 2024; Xu et al., 2024; Ye et al., 2025; Xu et al., 2025; Peng et al., 2025; Loveland et al., 2025) in the recommendation field have explored a similar idea: whether the performance of recommendation models can increase with parameter scaling. (Ardalani et al., 2022) explores the scaling properties of recommendation models across data, compute, and parameters, revealing that parameter scaling has limited impact on performance compared to data scaling, and highlights the need for future advancements in model architecture and system design. (Guo et al., 2024) identifies the embedding collapse phenomenon as the inhibition of scalability, where the embedding matrix leans towards occupying a low-dimensional subspace. It demonstrates a two-sided effect of feature interaction in recommendation models, with collapsed embeddings restricting learning and worsening the collapse.

### B.2 Efficient Recommendation.

How to set the dimension of embeddings is a common question for almost everyone studying collaborative filtering. This is because one of the core aspects of collaborative filtering is representing users or items with embeddings, and the quality of these embeddings directly determines the performance of the recommendation system. Reflecting on the efforts of recent years, to determine the optimal embedding dimension, many researchers have learned the embedding dimension automatically from the perspectives of neural architecture search (Liu et al., 2020; Qu et al., 2023), weight pruning (Wang et al., 2024; Luo et al., 2024; Liu et al., 2021; Tran et al., 2025) or exploring mixed dimension (Ginart et al., 2021). These methods seem to have

yielded notable results. The implicit assumption of these methods is that there exists an optimal dimension, meaning the performance exhibits a single-peak phenomenon.

### B.3 Denoising Recommendation

In recommendation systems based on implicit feedback, user behaviors (e.g., clicks, views) often contain *noisy interactions*—signals that do not truly reflect user preferences, such as clicks driven by item position or popularity rather than genuine interest Chen et al. (2025); Yang et al. (2024; 2025). These biases degrade model generalization and make learning reliable user preferences challenging. One of the simplest and most practical approaches is the drop-based method, which identifies noisy samples based on their loss values: samples with higher loss are more likely to be noisy, while those with lower loss are considered clean. drop-based method filters out noisy interactions before training. For example, T-CE (Wang et al., 2021) finds that noisy samples tend to have higher loss values, which can be used as a denoising signal. AutoDenoise (Lin et al., 2023) models denoising as a search-and-decision process and applies reinforcement learning to automate it. DCF (He et al., 2024) further improves performance by distinguishing between noisy interactions and hard samples, and by relabeling the noisy ones.

## C  Theoretical Discussions

### C.1  Proof of Theorem 1

*Proof.* Since $\Theta_0$ minimizes the clean objective, we have the first-order optimality condition:

$$\nabla_{\Theta}\mathbb{E}_{x\sim\mathcal{D}_0}[\ell(\Theta_0;x)] = 0. \tag{17}$$

Let us denote the noisy objective as:

$$\mathcal{L}_{\delta}(\Theta) := \mathbb{E}_{x\sim\mathcal{D}_{\delta}}[\ell(\Theta;x)] = (1-\delta)\mathbb{E}_{x\sim\mathcal{D}_0}[\ell(\Theta;x)] + \delta\mathbb{E}_{x\sim\mathcal{N}}[\ell(\Theta;x)]. \tag{18}$$

Then the gradient of the noisy objective at the clean optimum $\Theta_0$ is:

$$\begin{aligned}\nabla_{\Theta}\mathcal{L}_{\delta}(\Theta_0) &= (1-\delta)\nabla_{\Theta}\mathbb{E}_{x\sim\mathcal{D}_0}[\ell(\Theta_0;x)] + \delta\nabla_{\Theta}\mathbb{E}_{x\sim\mathcal{N}}[\ell(\Theta_0;x)]\\ &= \delta\cdot\Delta_{\text{noise}}.\end{aligned}$$

Assume the loss is twice continuously differentiable and the clean loss is locally $\mu$-strongly convex. Let $H := \nabla_{\Theta}^2\mathbb{E}_{x\sim\mathcal{D}_0}[\ell(\Theta;x)]|_{\Theta=\Theta_0} \succ 0$ be the positive definite Hessian. Then, using Taylor expansion:

$$\nabla_{\Theta}\mathcal{L}_{\delta}(\Theta_{\delta}) \approx \nabla_{\Theta}\mathcal{L}_{\delta}(\Theta_0) + H(\Theta_{\delta} - \Theta_0) + R, \tag{19}$$

where $R \in \mathcal{O}(\|\Theta_{\delta} - \Theta_0\|^2)$.

At the noisy optimum, the gradient vanishes:

$$0 = \nabla_{\Theta}\mathcal{L}_{\delta}(\Theta_{\delta}) \Rightarrow \Theta_{\delta} - \Theta_0 \approx -H^{-1}(\delta\cdot\Delta_{\text{noise}}). \tag{20}$$

Now compute:

$$\begin{aligned}Q(\Theta_{\delta}) &= \|\Theta_{\delta} - \Theta_0\|_F^2\\ &\approx \|\delta H^{-1}\Delta_{\text{noise}}\|_F^2\\ &= \delta^2\cdot\|H^{-1}\Delta_{\text{noise}}\|_F^2 + o(\delta^2).\end{aligned}$$

### C.2  Proof of Theorem 2

*Proof.* Let $f(u,i) = \text{MLP}([\mathbf{p}_u;\mathbf{q}_i]) + \mathbf{p}_u^{\top}\mathbf{q}_i$, and define the loss $\ell(\Theta;x) := \ell(f(u,i),y)$ for a sample $x = (u,i,y)$. The gradient of the loss with respect to model parameters $\Theta$ satisfies the chain rule:

$$\nabla_{\Theta}\ell(\Theta;x) = \nabla_z\ell(f(u,i),y)\cdot\nabla_{\Theta}f(u,i), \tag{21}$$

where $z := f(u, i) \in \mathbb{R}$ is the scalar output score.

Let $J_x := \nabla_\Theta f(u, i)$ denote the Jacobian of the scoring function. Taking norms on both sides yields:

$$\|\nabla_\Theta \ell(\Theta; x)\| \leq \|\nabla_z \ell(f(u, i), y)\| \cdot \|J_x\|. \tag{22}$$

We now analyze $\|J_x\|$. The scoring function can be decomposed as:

$$f(u, i) = \mathrm{MLP}([\mathbf{p}_u; \mathbf{q}_i]) + \mathbf{p}_u^\top \mathbf{q}_i. \tag{23}$$

The Jacobian of the inner product term is linear in $\mathbf{p}_u$ and $\mathbf{q}_i$, and contributes only a bounded term proportional to $\|\mathbf{p}_u\| + \|\mathbf{q}_i\|$. We focus on the MLP term, which dominates in depth.

Let the MLP be defined recursively as:

$$h^{(0)} = [\mathbf{p}_u; \mathbf{q}_i],$$
$$h^{(l)} = \sigma^{(l)}(W^{(l)} h^{(l-1)}), \quad \text{for } l = 1, \ldots, L,$$

where $\sigma^{(l)}$ is a pointwise activation function (e.g., ReLU or sigmoid), and $W^{(l)} \in \mathbb{R}^{d_l \times d_{l-1}}$ is the weight matrix at layer $l$. Then the gradient of the output with respect to parameters is:

$$\frac{\partial f(u, i)}{\partial \Theta} = \frac{\partial \mathrm{MLP}(h^{(0)})}{\partial \Theta} + \frac{\partial (\mathbf{p}_u^\top \mathbf{q}_i)}{\partial \Theta}. \tag{24}$$

Focusing on the MLP term, its Jacobian w.r.t. $h^{(0)}$ is:

$$\frac{\partial \mathrm{MLP}(h^{(0)})}{\partial h^{(0)}} = \prod_{l=1}^{L} D^{(l)} W^{(l)}, \tag{25}$$

where $D^{(l)} := \mathrm{diag}(\sigma'^{(l)}(W^{(l)} h^{(l-1)}))$ is a diagonal matrix of activation derivatives.

Taking operator norms, we get:

$$\left\| \frac{\partial \mathrm{MLP}(h^{(0)})}{\partial h^{(0)}} \right\| \leq \prod_{l=1}^{L} \|D^{(l)}\|_2 \cdot \|W^{(l)}\|_2. \tag{26}$$

Since $\|D^{(l)}\|_2 \leq \max_x |\sigma'^{(l)}(x)|$, which is bounded for standard activations like ReLU (bounded by 1) or sigmoid (bounded by $1/4$), we conclude:

$$\left\| \frac{\partial f(u, i)}{\partial \Theta} \right\| \leq C_\sigma \cdot \prod_{l=1}^{L} \|W^{(l)}\|_2, \tag{27}$$

for some constant $C_\sigma > 0$ depending on activation functions.

Combining this with the earlier chain rule, we obtain:

$$\|\nabla_\Theta \ell(\Theta; x)\| \leq \|\nabla_z \ell(f(u, i), y)\| \cdot C_\sigma \cdot \prod_{l=1}^{L} \|W^{(l)}\|_2. \tag{28}$$

Hence, the gradient norm can grow multiplicatively with the spectral norms of the MLP layers, implying exponential sensitivity in depth.

### C.3 Proof of Theorem 3

*Proof.* Decompose the embedding as $z_u = z_u^* + \delta_u$, where $z_u^* \in \mathcal{S}$ is the clean signal and $\delta_u \in \mathcal{S}^\perp$ is noise. The contrastive loss requires:

$$\cos(z_u', z_u'') = \frac{(z_u^* + \delta_u')^T(z_u^* + \delta_u'')}{\|z_u^* + \delta_u'\|\|z_u^* + \delta_u''\|} \approx 1. \tag{29}$$

Expanding this approximation:

$$1 - \frac{\|\delta_u' - \delta_u''\|^2}{2\|z_u^*\|^2} \approx 1 \implies \|\delta_u' - \delta_u''\|^2 \approx 0. \tag{30}$$

Since $\delta_u'$ and $\delta_u''$ are i.i.d., the expected variance is:

$$\mathbb{E}[\|\delta_u' - \delta_u''\|^2] = 2\mathbb{E}[\|\delta_u\|^2] - 2\mathbb{E}[\delta_u'^T \delta_u''] = 2\text{Var}(\delta_u). \tag{31}$$

Thus, $\text{Var}(\delta_u) \leq \epsilon^2$, constraining noise to low variance and ensuring embeddings primarily reside in $\mathcal{S}$.

### C.4 Derivation of Definition 4

*Derivation.* We begin by recursively applying the propagation rule described in Eq. (12). The embeddings of the $l$-th layer are given by:

$$\mathbf{P}^{(l)} = \tilde{A}\mathbf{P}^{(l-1)}, \tag{32}$$

where $\mathbf{P}^{(0)}$ represents the initial embeddings, and $\tilde{A}$ is the normalized adjacency matrix. This recurrence relation reflects the embedding propagation at each layer of the graph convolutional network.

The final embeddings after $L$ layers of propagation are obtained by taking the weighted sum across all layers:

$$\mathbf{P} = \sum_{l=0}^{L} \beta_l \tilde{A}^l \mathbf{P}^{(0)}, \tag{33}$$

where $\beta_l = \frac{1}{L+1}$ is the weight for the $l$-th layer.

Substituting the spectral decomposition of $\tilde{A}$ into this expression, we get:

$$\tilde{A} = U\Lambda U^T, \tag{34}$$

where $U$ is the matrix of eigenvectors and $\Lambda$ is the diagonal matrix of eigenvalues of $\tilde{A}$. Therefore, we can rewrite the embeddings as:

$$\mathbf{P} = \sum_{l=0}^{L} \beta_l (U\Lambda^l U^T)\mathbf{P}^{(0)}. \tag{35}$$

Using the associativity of matrix multiplication, this simplifies to:

$$\mathbf{P} = U\left(\sum_{l=0}^{L} \beta_l \Lambda^l\right) U^T \mathbf{P}^{(0)}. \tag{36}$$

Now, let us analyze the behavior of the filter coefficients $\sum_{l=0}^{L} \beta_l \Lambda^l$ in the spectral domain. For each eigenvalue $\lambda_k$ of $\tilde{A}$, the corresponding coefficient behaves as follows:

For $\lambda_k \approx 1$ (low-frequency components), we have:

$$\sum_{l=0}^{L} \beta_l \lambda_k^l \approx \sum_{l=0}^{L} \frac{1}{L+1} \cdot 1^l = 1. \tag{37}$$

This means that low-frequency components are preserved.

For $\lambda_k \approx 0$ (high-frequency components), we have:

$$\sum_{l=0}^{L} \beta_l \lambda_k^l \approx \sum_{l=0}^{L} \frac{1}{L+1} \cdot 0^l = 0. \tag{38}$$

This means that high-frequency components are attenuated, effectively suppressing noise.

Thus, the filter $\sum_{l=0}^{L} \beta_l \Lambda^l$ acts as a low-pass filter that suppresses high-frequency components and retains the low-frequency components of the embeddings.

As a result, the final embedding matrix $\mathbf{P}$ is dominated by the low-frequency components, and the rank of $\mathbf{P}$ is bounded by the number of significant eigenvalues (those corresponding to low-frequency components). Specifically, if $r$ is the number of significant eigenvalues, we can conclude that the rank of $\mathbf{P}$ is bounded by $r$:

$$\text{rank}(\mathbf{P}) \leq r. \tag{39}$$

This establishes the low-rank property of the embeddings after $L$ layers of propagation in the LightGCN model.

## D    Experimental Settings

### D.1    Hyperparameters

We pick the most commonly used Top-$k$ metrics NDCG@$k$ to measure performance, as well as for a comprehensive view of the phenomenon, we set $k$ to be 20. We set the embedding dimensions as powers of 2, starting from 2 and going up to $2^{16}$. We train the model for 300 epochs with a batch size of 2,048 using the Adam optimizer. The learning rate is fixed at 0.001 without weight decay (L2 regularization), and we disable gradient norm clipping. For negative sampling during training, we adopt a static strategy with 1 negative item per positive interaction, sampled uniformly from the item space. We evaluate the model after every training epoch and implement early stopping if validation performance does not improve for 10 consecutive epochs. All models operate in essentially the same way; apart from differences in architecture, aspects like training loss are nearly identical.

### D.2    Pytorch Code of BPR_Drop

Code of BPR_Drop

```python
class BPRdropLoss(nn.Module):
    """BPRLoss, based on Bayesian Personalized Ranking

    Args:
        - gamma(float): Small value to avoid division by zero

    Shape:
        - Pos_score: (N)
        - Neg_score: (N), same shape as the Pos_score
        - Output: scalar.

    Examples::

        >>> loss = BPRLoss()
        >>> pos_score = torch.randn(3, requires_grad=True)
        >>> neg_score = torch.randn(3, requires_grad=True)
        >>> output = loss(pos_score, neg_score)
        >>> output.backward()
    """
```

```python
def __init__(self, save_ratio, get_low, gamma=1e-10):
    super(BPRdropLoss, self).__init__()
    self.gamma = gamma
    self.save_ratio = save_ratio
    self.get_low = get_low

def forward(self, pos_score, neg_score):
    # loss = -torch.log(self.gamma + torch.sigmoid(pos_score - neg_score)).mean()
    per_sample_losses = -torch.log(self.gamma + torch.sigmoid(pos_score - neg_score))

    num_samples = per_sample_losses.size(0)
    k = int(num_samples * self.save_ratio)

    k = max(k, 1)

    if self.get_low == 1:
        _, indices = torch.topk(per_sample_losses, k, largest=False)
    else:
        _, indices = torch.topk(per_sample_losses, k, largest=True)

    selected_losses = per_sample_losses[indices].mean()
    return selected_losses
```

### D.3 Detailed description of the datasets

In our research, we utilized four distinct datasets to ensure the comprehensiveness and robustness of our analysis. Each dataset contributes unique characteristics to our study. Table 3 summarizes the statistics of the 10 datasets used in our experiments, demonstrating the comprehensiveness of our evaluation. The datasets span a wide range of interaction scales, from Small (e.g., ML-100K) to X-Large (e.g., Amazon_Books), and cover various sparsity levels, from relatively dense (e.g., ML-1M) to extremely sparse (e.g., Amazon_Beauty). Notably, the majority of datasets fall into the Ultra-sparse category, reflecting the real-world challenge of data sparsity in recommendation. This diversity ensures a thorough and robust evaluation of model performance across different data regimes. The color-coded annotations provide intuitive insights into dataset characteristics.

### D.4 Reason for Choosing Models

BPR (Rendle et al., 2009) is a classic collaborative filtering model known for its simplicity and ease of use. Many subsequent influential methods have been developed based on it. NeuMF (He et al., 2017), in contrast, is a highly representative deep learning-based model that adopts a classic two-tower architecture, combining linear modeling via matrix factorization with nonlinear interaction modeling through neural networks. As research progressed, it became clear that the relationships between users and items can be naturally modeled as graph structures. This insight led to the emergence of numerous graph-based collaborative filtering approaches, such as LightGCN (He et al., 2020). Building upon graph structures, graph self-supervised models—such as SGL (Wu et al., 2021)—further enhance performance by leveraging both feature and structural information, attracting significant attention. These are the reasons why we chose to focus on these three categories of models. The more specific information is at Table 2.

## E Experiments

### E.1 All Observations

we present additional experimental results on various datasets. As shown in Figures 4, 5, and 6, SGL demonstrates a slower performance degradation trend compared to LightGCN, which in turn outperforms NeuMF, while BPR performs similarly to SGL. These results are consistent with our theoretical analysis,

Table 2: collaborative filtering models and their publication details

| Model | Paper Title | Year | Venue | Citation |
|-------|-------------|------|-------|----------|
| BPR | BPR: Bayesian Personalized Ranking from Implicit Feedback | 2009 | UAI | 8,232 |
| NeuMF | Neural Collaborative Filtering | 2017 | WWW | 8,052 |
| LightGCN | LightGCN: Simplifying and Powering Graph Convolution Network for Recommendation | 2020 | SIGIR | 4,358 |
| SGL | Self-supervised Graph Learning for Recommendation | 2021 | SIGIR | 1,423 |

Table 3: Statistics of the 10 benchmark datasets used in our experiments, highlighting both the dataset scale and sparsity level. **Scale** is categorized based on the number of interactions: Small (<1M), Medium (1M–5M), Large (5M–10M), and X-Large (>10M). **Sparsity Level** is categorized based on the sparsity ratio: Dense (<0.6), Semi-sparse (0.6–0.95), Sparse (0.95–0.99), and Ultra-sparse ($\geq$0.99). Colored backgrounds provide intuitive cues for comparison across datasets.

| Dataset | #Users | #Items | #Interactions | Sparsity | Scale | Sparsity Level |
|---------|--------|--------|---------------|----------|-------|----------------|
| ML-100K | 943 | 1,682 | 100,000 | 0.93695 | Small | Semi-sparse |
| Modcloth | 9,000 | 10,000 | 500,000 | 0.99945 | Small | Ultra-sparse |
| Amazon_Baby | 531,890 | 64,426 | 915,446 | 0.99997 | Small | Ultra-sparse |
| ML-1M | 6,040 | 3,706 | 1,000,209 | 0.97570 | Medium | Sparse |
| Douban | 11,257 | 2,281 | 1,330,000 | 0.99945 | Medium | Ultra-sparse |
| Pinterest | 55,187 | 9,911 | 1,445,622 | 0.99736 | Medium | Ultra-sparse |
| Amazon_Beauty | 1,210,271 | 249,274 | 2,023,070 | 0.99999 | Medium | Ultra-sparse |
| Yelp | 1,326,101 | 174,567 | 5,261,669 | 0.99998 | Large | Ultra-sparse |
| Gowalla | 107,092 | 1,280,969 | 6,442,892 | 0.99990 | Large | Ultra-sparse |
| Amazon_Books | 8,026,324 | 2,330,066 | 22,507,155 | 0.99990 | X-Large | Ultra-sparse |

confirming the validity of our investigation into the noise robustness of different models. Additionally, we observed a double-peak phenomenon in other datasets as well (e.g., Figure 4 (b, e, h)), along with an logarithm phenomenon (e.g., Figure 4 (j, k, l)). Moreover, the single-peak phenomenon widely discussed in prior work is also clearly present in our experiments (e.g., Figure 5 (a, b, c, etc.)). We argue that SGL's ability to maintain or even improve performance as the embedding dimension increases is largely due to the regularization and robustness enhancement effects introduced by contrastive learning. In contrast, LightGCN lacks auxiliary training signals and regularization methods beyond supervised learning, making it more prone to overfitting or performance degradation in high-dimensional settings. Overall, these experimental results further validate the generality of the newly observed phenomena and confirm that the observed performance trends are well aligned with our theoretical analysis.

## E.2 Effect of Varies Dropping Ratio

The Figures 7 show the NDCG@20 of BPR_Drop with different save-ratios and standard BPR on ML -100K and Douban datasets as embedding size varies. In Figure (a) for ML-100K, BPR_Drop with save - ratios 0.95, 0.9, 0.85, 0.8 and BPR are compared as embedding size ranges from $2^9$ to $2^{15}$. BPR_Drop exhibits distinct trends, indicating sensitivity to scaling. In Figure (b) for Douban, similar comparisons are made. BPR_Drop's performance curves show unique patterns with embedding size changes, suggesting better scalability than standard BPR. These plots offer insights into BPR_Drop's scaling behavior for optimization in different datasets.

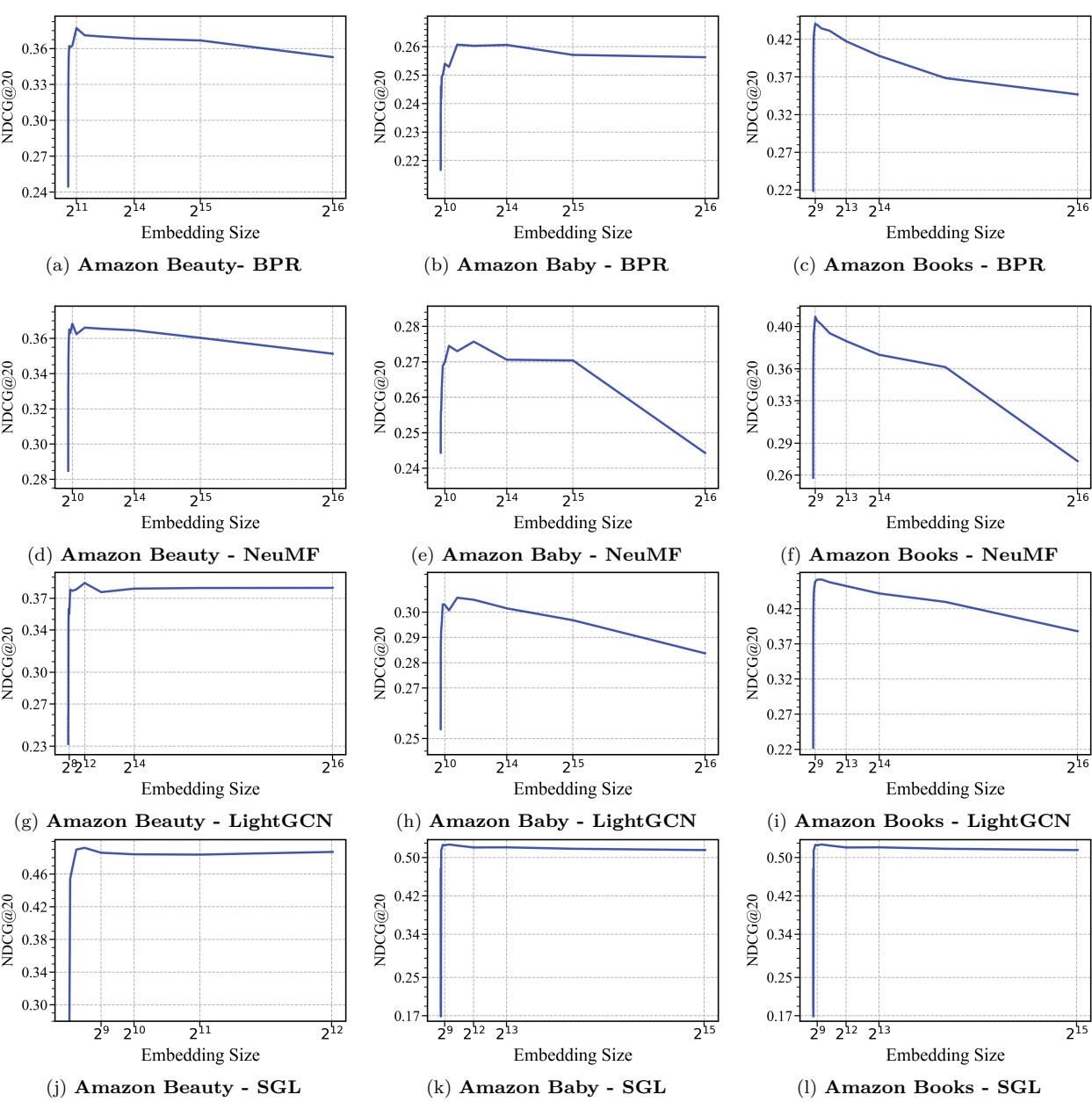

Figure 4: Scale the embedding dimension exponentially by a factor of 2 across different collaborative filtering models and datasets. Each row corresponds to a model (BPR, LightGCN, SGL, NeuMF), and each column represents a dataset (Amazon Beauty, Amazon Baby, Amazon Books).

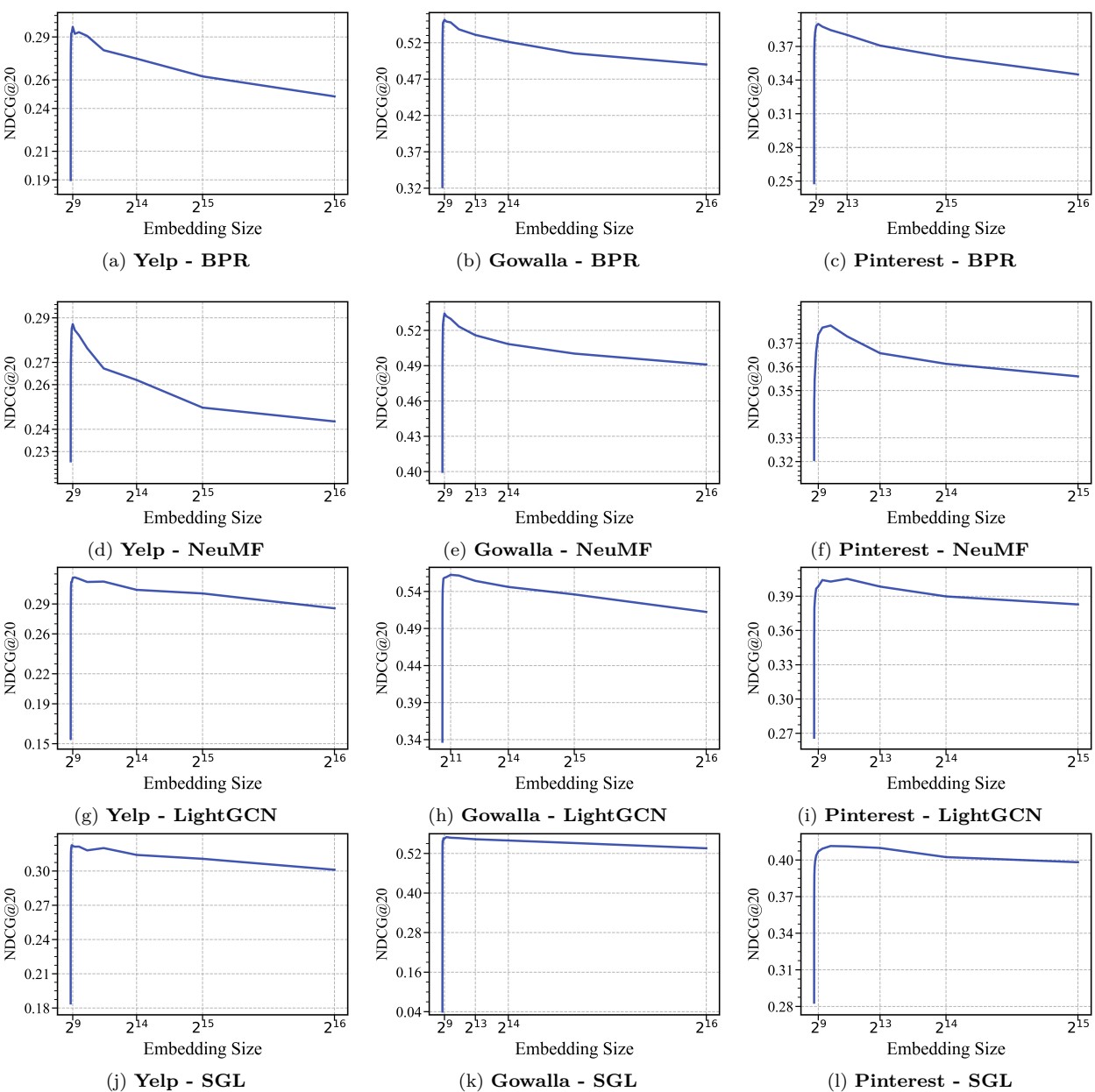

Figure 5: Scale the embedding dimension exponentially by a factor of 2 across different collaborative filtering models and datasets. Each row corresponds to a model (BPR, LightGCN, SGL, NeuMF), and each column represents a dataset (Yelp, Gowalla, Pinterest).

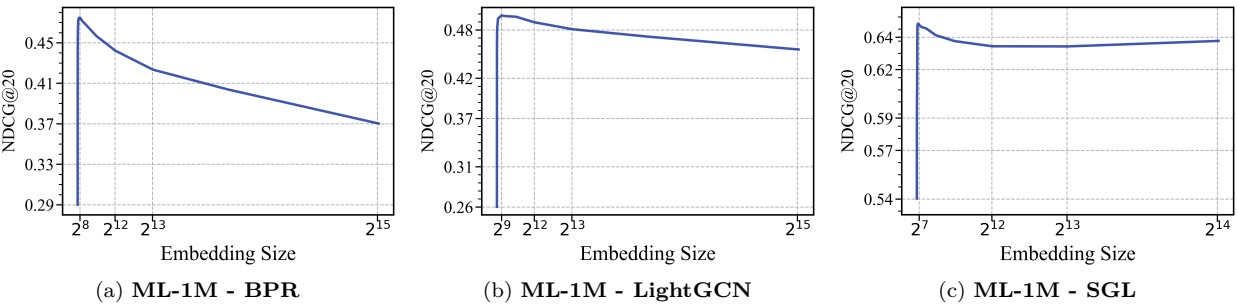

Figure 6: Scale the embedding dimension exponentially by a factor of 2 across different collaborative filtering models and datasets.

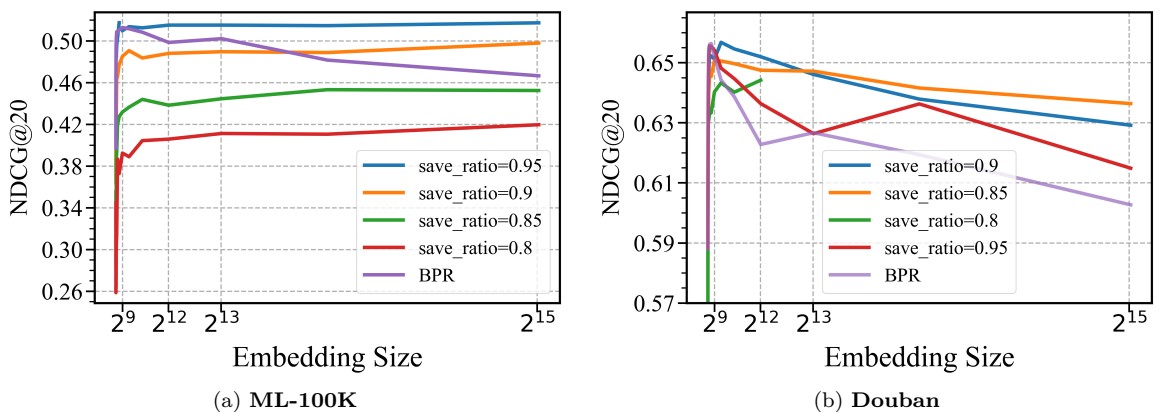

Figure 7: Comparing the standard BPR with the denoising strategy-based BPR_Drop in ML-100K and Douban, we can clearly observe that the performance degradation has significantly improved.

