# OpenReview forum: "Understanding Embedding Scaling in Collaborative Filtering"
_TMLR — Accepted by TMLR_

### Review · Reviewer_nKt4 · 2025-10-20

**Summary Of Contributions:**

The paper focuses on giving reason to a popular question in collaborative filtering "Why does expanding embedding dimensions not always result in performance enhancements". The paper evaluates 4 different models across 10 different datasets of varying sparsity levels. The observe two novel phenomenons double-peak and logarithmic. For the former, as the embedding dimension increases the performance improves, then declines, rises again and eventually drops. For the latter, it follows a perfect logarithmic curve. The paper also tries to explain the cause of double-peak phenomenon, links it to noise and sparsity and finally theoretically analyze the noise robustness of collaborative filtering models.

## Strengths
- The paper tackles a relevant question of "why expanding embedding dimension not always results in performance enhancements in collaborative filtering".
- They use sufficient datasets and do sufficient experiments.
- Testing and comparison of models BPR and BPR_Drop justifies the dependence of double peak phenomenon on noise in the datasets, which is a nice addition.
- The theoretical analysis of noise does gives some intuition as to why some models consistently perform better than others.

## Weakness
I am mentioning the weakness and my questions in the *Requested Changes* section, please see that.

**Audience:**

Yes

**Audience Explanation:**

Yes the paper target relevant questions and some people will definitely be interested in it.

**Claims And Evidence:**

No

**Claims Explanation:**

I have some doubts about the claims and need more convincing, I am posting my queries in requested changes.

**Requested Changes:**

## Weakness/Questions
I have marked question as [Critical] which are critical to secure my recommendation for acceptance.

1. [Critical] First a basic question, I see authors call the logarithmic plot ideal and would want to achieve it, can you convince me? Based on figure 1 it looks like double-peak gives better results using fewer embedding dimensions and if one finds the first peak it's a better result as it gives better performance using smaller embedding dimension.
2. [Critical] Extending to the first question, I understand that for a logarithmic curve finding the embedding dimension will be much easier, however I see that in double-peak second peak is always as lower accuracy than the first peak, so my objective will be to find the first peak, which should not be that hard. Is there an example where the second peak is higher than the first one?
3. [Critical] I fail to understand the reason why increasing the embedding dimension will shift the model's learning from phase 2 to phase 3, i.e., learning noise which degrades performance to adapting to the noise thereby improving the performance? This is very core question for this paper and is not clear to me after reading the draft, I request the authors to explain it clearly.
4. [Critical] Authors compare the phenomenon of double-peak with sparse double descent by linking change in embedding dimension with sparsity, does it mean that double-peak is just sparse double descent with changing embedding dimension? How is it different from sparse double descent?
5. [Critical] In the experiments authors use early stopping, which won't allow the model to overfit to the training data, so the numbers in plot are not numbers for overfitted model. From my understanding the models should be trained till training error reduces so that the plots will also show overfitting as the embedding dimensions blows up.
6. In first paragraph of page 3, line 2-3 there seem to be incorrect notations use, for $P_u \in \mathbb{R}^k$ there is no $i$ and similarly for $q_i \in \mathbb{R}^k$ there is no $j$ used.

---

> ### Author Response · Authors · 2025-10-26
>
> We sincerely thank the reviewer for taking the time to rigorously and carefully review our work and provide detailed feedback. We will respond to the reviewer’s comments point by point, aiming for clarity and addressing concerns.
>
> ---
>
> **For Q1**: The logarithmic trend is ideal because it provides stable, predictable performance gains as embedding size increases. Indeed, we agree that small embedding sizes can sometimes achieve good performance. And, we would like to emphasize that the contribution of this work lies in the first discovery of these two phenomena, and it is these observations that sparked subsequent discussions.
>
> **For Q2**: We appreciate this insightful question. After thorough analysis of all experimental results, we confirm that in our observed double-peak phenomena, the second peak is consistently lower than the first peak, as you said. There is no case where the second peak surpasses the first. We want to clarify that the significance of the double-peak phenomena lies in its revelation of noisy interaction effects, which help us understand scaling limitations and develop strategies (e.g., denoising) to achieve more robust performance. Discovering and understanding new phenomena is the core of this work.
>
> **For Q3**: The four phases are as follows: the model first tends to learn clean samples (Phase 1, improves), then memorizes noisy samples (Phase 2, declines) [1]. As the embedding size increases, the enlarged space dilutes the negative impact of noise (Phase 3, rebounds); finally, an excessively large embedding size leads to overfitting (Phase 4, drops).
>
> **For Q4**: Yes, your understanding is generally correct. The two phenomena exhibit similar curves because they share the same underlying causes—sparsity and data noise. It was precisely this similarity in curve patterns that led us to analyze the mechanism behind the double-peak phenomenon. There are also key differences. Specifically, sparse double descent is discussed in the general machine learning context, where label noise and model pruning are involved. In contrast, our double-peak phenomenon arises in collaborative filtering when scaling the embedding dimension under noisy interactions.
>
> **For Q5**: We use early stopping to ensure a fair and optimal comparison of generalization performance across all embedding dimensions. It is a standard practice to select the best model checkpoint. Crucially, the reported phenomena are not an artifact of the training procedure. And, in general, a decrease in training error does not necessarily indicate a reduction in test error. All our observations are made at the point of best test performance, which aligns with practical expectations in machine learning—our focus is always on improving test performance rather than training performance.
>
> **For Q6**: Thanks for the thorough review — that was a typo and we have corrected it immediately.
>
> [1] Arpit, D., Jastrzębski, S., Ballas, N., Krueger, D., Bengio, E., Kanwal, M.S., Maharaj, T., Fischer, A., Courville, A., Bengio, Y. and Lacoste-Julien, S., 2017, July. A closer look at memorization in deep networks. In International conference on machine learning (pp. 233-242). PMLR.
>
> ---
>
> We sincerely thank you for the time and effort you devoted to reviewing our work. We greatly appreciate it, and welcome any further questions or discussion!

---

> ### Author Response · Authors · 2025-11-11
>
> Dear Reviewer,
>
> I completely understand that you may be very busy right now. As the two-week discussion deadline is approaching, I’d really appreciate it if you could take a moment to check whether there are any remaining questions we should discuss. We will make every effort to address your concerns. Thank you very much for your time and effort!
>
> All best, \
> Authors

---

### Review · Reviewer_GSnC · 2025-10-27

**Summary Of Contributions:**

The contributions of this work include (1) two new phenomena i.e., "double peaks" and "logarithmic" discovered from extensive experiments; (2) uncovering that interacting noise is the main reason for double peaks; (3) analysis of the model robustness of  four typical collaborative filtering.

**Audience:**

Yes

**Audience Explanation:**

This work may be interesting to a large number of potential readers including theoretical researchers and engineers in recommandation systems. It's empirical findings (double peaks and logarithmic) and theoretical analysis are new.

**Broader Impact Concerns:**

To the best of my knowledge, there are no broader impact concerns.

**Claims And Evidence:**

Yes

**Claims Explanation:**

This work is important because it disgrees with the old understanding that "there is a unique optimal embedding dimension". It for the first time show two new findings (double peaks and logarithmic), and shows us the cause (interactive noises). It also theoretically explain why noise-resilient structures like SGL has better scalability.

**Requested Changes:**

This paper is well motivated and very interesting. I only have very minor comments as follows.

1. The writing is not always satisfactory. There are typos like "two novel phenomenon".

2. I think it is important for the experiments to be repeated. I suggest the authors opensource the code for reproductible evaluation.

---

> ### Author Response · Authors · 2025-10-27
>
> Thank you for your very positive recognition of our work! Being able to help both researchers and engineers is our greatest hope, and we sincerely appreciate it. Thank you very much!
>
> ----
>
> **For Q1**: We have thoroughly reviewed the manuscript to ensure there are no typos.
>
> **For Q2**: This aligns with our intention — we will release not only the code but also the complete experimental records, ensuring everyone has access to the raw data.
>
> ---
>
> Once again, thank you for your strong recognition of our work; we truly appreciate it!

---

> ### Comment · Reviewer_GSnC · 2025-11-11
>
> The revision addresses my concerns, and I have no further questions.

---

### Review · Reviewer_BBUR · 2025-10-30

**Summary Of Contributions:**

This work investigates how embedding dimension scaling affects the performance of collaborative filtering models. Through experiments on 10 popular datasets and 4 classical models (BPR, NeuMF, LightGCN, SGL), the authors discover two new phenomena: the double-peak effect, where performance swing with increasing dimensions, and the logarithmic effect, where performance grows in a logarithmic manner. They attribute these effects to differences in noise robustness. Theoretical analysis reveals that BPR and NeuMF suffer from gradient instability, while graph-based models remain robust through structural mechanisms like spectral filtering and contrastive regularization. Empirical results and a denoising strategy futher show that noisy interactions cause the double-peak phenomenon.

Strength:
1) This work is written with great clarity, rigor, and interest. It builds insightful observations from newly discovered phenomena and tightly connects theoretical analysis with experimental validation. The overall logic is coherent and well-structured, leaving little room for criticism.
2) This work provides a deeper understanding of embedding scalability and offering insights into noise robustness and provides formal proofs to link noise robustness to model design.
3) This work includes a comprehensive set of experiments, demonstrating substantial implementation effort. This makes the work particularly solid and convincing.

**Audience:**

Yes

**Audience Explanation:**

I think the observations in this paper will capture the interest of people in this field.

**Claims And Evidence:**

Yes

**Claims Explanation:**

The important new phenomena discovered in this work will serve as one of the key contributions to the future development of generative recommendation. It shifts the scaling perspective from improving Transformer architecture to emphasizing the intrinsic characteristics of collaborative filtering—an insight of substantial significance.

**Requested Changes:**

Weakness:
1) While the proposed Double-peak and Logarithmic Phenomena offer useful insights, they may not fully explain all experimental observations. For example, in Figures 5 and 6, some cases still show a single-peak behavior, which the current framework does not entirely account for.
2) Some typos, such as incorrect singular or plural usage of phenomenon in the abstract.

---

> ### Author Response · Authors · 2025-10-31
>
> Thank you very much for your highly positive feedback! We will respond to your reviews point by point.
>
> ----
>
> **For Q1**: We appreciate your review of Figures 5–6 and agree that the single-peak pattern is more common in some datasets. While previous studies focused on the single-peak pattern—which indeed exists—our work highlights two newly identified phenomena: the double-peak and logarithmic patterns.
>
> **For Q2**: We have promptly corrected the typo and carefully checked the other parts ourselves.
>
> ---
>
> We again thank you for recognizing the many highlights of our work!

---

> > ### Comment · Reviewer_BBUR · 2025-11-11
> >
> > The revision addresses my concerns, and I have not further questions.

---

### Decision · Action_Editor_pvt3 · 2025-12-08

**Recommendation:** Accept with minor revision

**Audience:**

Yes

**Audience Explanation:**

This paper will attract some audience in collaborative filtering and recommendations.

**Claims And Evidence:**

Yes

**Claims Explanation:**

This paper provides a rigorous analysis of why increasing embedding dimensions does not always improve collaborative filtering performance. The empirical study across 10 datasets and 4 models reveals consistent double-peak and logarithmic phenomena, and the theoretical explanation through noise robustness is valuable. The work is well written and experimentally thorough.
The revised version should clearly explain why the logarithmic behavior is viewed as ideal, clarify the mechanism behind the phase transition in the double-peak effect, discuss its relationship to sparse double descent, and fix minor notation issues. Overall, reviewers are positive about the contribution, so I recommend acceptance with minor revisions.